# Genome-Wide Screening in Human Embryonic Stem Cells Highlights the Hippo Signaling Pathway as Granting Synthetic Viability in ATM Deficiency

**DOI:** 10.3390/cells12111503

**Published:** 2023-05-29

**Authors:** Ruth Viner-Breuer, Tamar Golan-Lev, Nissim Benvenisty, Michal Goldberg

**Affiliations:** 1The Azrieli Center for Stem Cells and Genetic Research, The Hebrew University, Givat-Ram, Jerusalem 9190401, Israel; ruth.viner@mail.huji.ac.il (R.V.-B.); tamar.gl@mail.huji.ac.il (T.G.-L.); 2Department of Genetics, Institute of Life Sciences, The Hebrew University, Givat-Ram, Jerusalem 9190401, Israel

**Keywords:** *ATM*, Hippo signaling pathway, ataxia–telangiectasia, DNA damage, neurodegeneration, human embryonic stem cells, CRISPR loss-of-function library

## Abstract

*ATM* depletion is associated with the multisystemic neurodegenerative syndrome ataxia–telangiectasia (A–T). The exact linkage between neurodegeneration and *ATM* deficiency has not been established yet, and no treatment is currently available. In this study, we aimed to identify synthetic viable genes in *ATM* deficiency to highlight potential targets for the treatment of neurodegeneration in A–T. We inhibited ATM kinase activity using the background of a genome-wide haploid pluripotent CRISPR/Cas9 loss-of-function library and examined which mutations confer a growth advantage on *ATM*-deficient cells specifically. Pathway enrichment analysis of the results revealed the Hippo signaling pathway as a major negative regulator of cellular growth upon ATM inhibition. Indeed, genetic perturbation of the Hippo pathway genes *SAV1* and *NF2*, as well as chemical inhibition of this pathway, specifically promoted the growth of *ATM*-knockout cells. This effect was demonstrated in both human embryonic stem cells and neural progenitor cells. Therefore, we suggest the Hippo pathway as a candidate target for the treatment of the devastating cerebellar atrophy associated with A–T. In addition to the Hippo pathway, our work points out additional genes, such as the apoptotic regulator *BAG6*, as synthetic viable with *ATM*-deficiency. These genes may help to develop drugs for the treatment of A–T patients as well as to define biomarkers for resistance to *ATM* inhibition-based chemotherapies and to gain new insights into the *ATM* genetic network.

## 1. Introduction

The ataxia–telangiectasia mutated (ATM) gene encodes a well-characterized serin/threonine protein kinase belonging to the phosphoinositide 3-kinase-related kinase (PIKK) family. ATM targets numerous proteins, some of which are shared with related PIKKs, such as ATR and DNA-PK [1]. ATM plays a variety of roles in maintaining cell homeostasis. It is mainly known as the key kinase regulating the cellular response to DNA double-strand breaks (DSBs), acting as a DNA damage transducer, and regulating checkpoint activation, DNA repair, and apoptosis [2]. In unstressed cells, ATM is present as an inactive homodimer, and upon the induction of DSBs, the MRE11-RAD50-NBS1 (MRN) complex rapidly recruits ATM to the damage sites, where it is monomerized and activated [3]. As ATM becomes active, it phosphorylates multiple downstream targets, including ATM itself; the histone variant H2AX, which stimulates the DNA damage response (DDR) cascade; and CHEK2, which activates DNA damage checkpoints. Other downstream targets of ATM include additional key players in the DDR process, such as BRCA1, 53BP1, and p53. ATM also plays a role in telomere maintenance and it acts as a neuronal epigenetic regulator [1,4]. Apart from its significant nuclear role in safeguarding genome integrity, accumulating evidence reveals a complex cytoplasmic role for ATM. In the cytosol, ATM is activated by oxidative stress and regulates various processes, such as mitochondrial reactive oxygen species (ROS) production, protein quality control, removal of damaged mitochondria and peroxisomes, angiogenesis, and glucose metabolism. ATM dimerizes in the cytosol, a process required for its cytoplasmic functions [5,6,7].

The extent to which ATM is involved in maintaining cellular homeostasis differs between cell types and developmental stages. As an example, human neural progenitor cells (NPCs) and differentiated neurons mostly depend on ATM for DSB repair, while human embryonic stem cells (hESCs) rely predominantly on ATR [8]. Accordingly, ATM is upregulated during neuronal differentiation, whereas ATR is downregulated [9].

Homozygous mutations in the ATM gene are associated with a rare multisystemic syndrome known as ataxia–telangiectasia (A–T). A–T patients suffer from a wide range of symptoms, the most devastating of which is progressive cerebellar atrophy resulting from the death of Purkinje cells in the cerebellum, leading to severe ataxia that manifests during early childhood. Additional symptoms include oculocutaneous telangiectasia, immunodeficiency, radiosensitivity, cancer predisposition, increased risk for metabolic diseases, and premature aging [4,10]. The current understanding of ATM function can explain most of these symptoms, but not all of them. The mechanisms behind the correlation between neurodegeneration and ATM depletion has yet to be solved, and currently no cure is available for halting or delaying the neurological deterioration [4].

The Hippo signaling pathway is a highly conserved cascade of tumor-suppressing kinases that regulates cell apoptosis. It is involved in various processes, such as cell fate determination, stem cell regulation, regeneration, and immunity [11,12]. In the canonical Hippo pathway in mammals, the MST1/2-SAV1 complex phosphorylates and activates the LATS1/2-MOB1A/B complex, which in turn phosphorylates the transcriptional co-activators YAP and TAZ, resulting in their cytoplasmic retention. Upon inhibition of the Hippo pathway, YAP/TAZ are translocated to the nucleus, where they interact with members of the TEAD family of transcription factors, which trigger transcription of proliferation, stemness, and cell growth genes [11,13]. The Hippo pathway is activated by various physical and chemical signals, such as DNA damage. DSB formation triggers Hippo signaling activation in an ATM-dependent manner [14,15]. Additional effectors, such as NF2, interact with Hippo pathway members and induce its activation. NF2 recruits the LATS1/2 complex to the plasma membrane, where it is phosphorylated and activated [11].

CRISPR-based loss-of-function screens have benefited genetic research over the past few years, revealing various functional and mechanistic insights. One phenomenon that can be investigated using such screens is synthetic lethality—a situation in which the simultaneous perturbation of two neutral genes results in a significant reduction in cell viability. The concept of synthetic lethality has gained a great deal of attention in the field of cancer drug development because it can be utilized to develop effective drug combinations for cocktail therapies as well as to identify background mutations that confer sensitivity to specific drugs [16,17]. On the other hand, synthetic viability is a phenomenon where a combination of mutations in two different genes can prevent the deleterious effect of a single mutation. Identifying synthetically viable interactions can be useful for finding biomarkers for cancer treatment resistance and for identifying targets for the rescue of cells with otherwise deleterious mutations [18].

Some major limitations of standard CRISPR screens are the low probability of obtaining homozygous mutations, which mask gene effects by intact alleles, as well as noise from background mutations, which is primarily found in cancer cells. In a previous work, we derived haploid hESCs from artificially activated human oocytes [19] and generated a CRISPR/Cas9 loss-of-function library targeting 18,166 protein-coding genes [20]. The haploid nature and the intact genome characterizing these cells address both above limitations. Using this library, we were able to map the essential genes for the growth of pluripotent cells. In this analysis, ATM mutation was found to have a mild negative effect on the growth of the cells already in the pluripotent state [20]. Here we utilize our established library to identify synthetic viable interactions with ATM loss-of-function that can reverse the negative effect of ATM mutation in order to identify potential targets for the treatment of neurodegeneration in A–T patients. These genes can also be used as biomarkers for resistance to pharmacological inhibition of ATM activity in cancer treatment. Our analysis highlights the Hippo signaling pathway as a synthetic viable pathway with ATM deficiency in hESCs. We demonstrate that depletion of Hippo pathway components, as well as chemical inhibition of the cascade, reverses the negative growth effect of ATM depletion in pluripotent cells. Moreover, we found that inhibition of the Hippo pathway increases the survival of ATM-depleted NPCs. Based on these results, we propose a novel therapeutic approach for ATM-dependent neurodegeneration, with the Hippo pathway as a potential target.

## 2. Materials and Methods

### 2.1. Cell Lines, Plasmids and Reagents

Diploid pES10 cells and the haploid hESCs cell line h-pES10 previously isolated by us [19], as well as hESC cells from the CRISPR/Cas9 library derived from h-pES10 (hereafter referred to as “library cells”) established in our lab [20], were used under the Israeli guidelines concerning hESC research (https://www.academy.ac.il/Index3/Entry.aspx?nodeId=842&entryId=18759 (accessed on 27 April 2023)). Mutant HEK293T cells were obtained from R. Weinberg (Whitehead Institute). LentiCRISPR v2 was a gift from Feng Zhang (Addgene, cat. No. 52961), and lentiCRISPRv2-neo was a gift from Brett Stringer (Addgene cat. no. 98292). Key resources and reagents are summarized in Appendix A.

### 2.2. Cell Culture

Feeder mouse embryonic fibroblasts (MEF cells) were seeded on gelatin (MP Biomedicals, Illkirch, France) in Dulbecco’s modified Eagle’s medium (DMEM, Sigma Aldrich, St. Lois, MO, USA) 3 h to 1 d prior to human ESC seeding. Human ESCs and library cells were cultured at 37 °C and 5% CO_2_ on previously prepared MEF plates in human ES medium containing knockout DMEM (ThermoFisher Scientific, Paisley, UK) supplemented with 15% knockout serum replacement (KSR, ThermoFisher Scientific), 0.1 mM nonessential amino acids, 2 mM L-glutamine (Biological Industries, BI, Kibbutz Beit Haemek, Israel), 50 µg/mL streptomycin (BI), 50 units/mL penicillin (BI), 0.1 mM β mercaptoethanol, and 8 ng/mL basic fibroblast growth factor (bFGF). Alternatively, during the ATMi screen experiment and for FACS or CellTiter-Glo assays, cells were seeded on Matrigel-coated plates (BD Biosciences, Le Pont-de-Claix, France), in mTeSR1 media (STEMCELL Technologies, Vancouver, BC, Canada) supplemented with 1:1000 ROCK inhibitor Y-27632 (ROCKi, Stemgent, Boston, MA, USA) in the absence of feeder cells. HEK293T cells were cultured at 37 °C and 5% CO2 in DMEM supplemented with 10% fetal bovine serum (BI), 2 mM L-glutamine, 50 units/mL penicillin, and 50 µg/mL streptomycin. Cells were detached using trypsin (BI) or TRYPLE select (ThermoFisher Scientific).

### 2.3. Western Blotting

Cells were lysed using 2XSDS + DTT lysis buffer, incubated at 100 °C for 15 min, and frozen at −20 °C. Before use, lysates were re-incubated at 100 °C for 5 min followed by a short spindown. Protein lysates were electrophoresed on 8% SDS polyacrylamide gels followed by overnight wet transfer at 4 °C to nitrocellulose membranes that were blocked with TBS-T buffer (0.15 M NaCl, 0.05 M Tris hydroxymethyl methylamine and 0.1% Tween-20, pH 7.4–7.6) containing 5% nonfat milk (BD) for 0.5 h at room temperature. Membranes were then washed with a TBS-T buffer and incubated overnight at 4 °C with primary antibodies, diluted according to manufacturer instruction in 4% BSA. Membranes were washed with TBS-T buffer and incubated with secondary antibody (1:5000 in TBS-T with 4% nonfat milk) for 2 h at room temperature, followed by three washes with TBS-T for 10 min each. Chemiluminescence was visualized with an ECL detection kit (Biogate, Ness Ziona, Israel) using gel imager (ChemiDoc, Bio Rad, Hercules, CA, USA or Fusion FX, Viber Lumart, Marne-la-Vallée, France) and quantified using the software program ImageJ version 1.51a (NIH). Results were normalized to loading control (GAPDH or Tubulin) and to no treatment sample. Full-length gel images are available in the Appendix A Western blot.

### 2.4. Growth Assays

Cells were plated in Matrigel-coated 96-well plates in mTeSR1 medium at a density of 10,000 cells per well. One day after seeding, cells were treated with a medium containing either the ATM inhibitor KU55933 (Sigma Aldrich, St. Lois, MO, USA), CGP3466B (Tocris, Bristol, UK), or Lats-IN-1 (LATSi, MCE, Monmouth Junction, NJ, USA) at indicated concentrations. After 1 h, Neocarzinostatin (NCS) (1:15,000, ~33 ng/mL) was added and left to rest for 1 h. The NCS was then deactivated by light exposure for 10 min. Twenty-four hours after initial ATMi administration, the medium was replaced again with fresh medium without drugs. Cell density was estimated at different time points using CellTiter-Glo luminescent cell-viability assay (Promega, Madison, WI, USA) according to the manufacturer’s instructions. Luminescence was recorded using a plate reader (Plate reader Synergy H1, Biotek, Winooski, VT, USA). Luminescence reads were normalized to no treatment control.

### 2.5. ATM Inhibition Screen

Library cells were plated in Matrigel-coated 10 cm plates (4–6 million cells per plate) in m-TeSR1 medium with 1:1000 ROCK inhibitor. Four conditions were tested in this assay: (1) no treatment (NT-ct); (2) DMSO 0.36% (D-ct); (3) ATMi 60 μM; and (4) ATMi 90 μM. Five plates were seeded per condition. Twenty-four hours post seeding, the compounds were added to all plates, except the NT-ct plates, where the cells were harvested and frozen. Treatment-containing medium was replaced with fresh medium 24 h post-treatment, allowing the cells to recover. D-ct cells were harvested and frozen at day 4, and the rest of the cells underwent a second round of treatment at day 6. A few days later, the cells were collected, and some were frozen and some taken for further growth. Two additional rounds of treatments were introduced to the remaining cells before final harvesting. The total duration of the procedure was about three weeks. DNA extraction, amplification, and sequencing were performed as described before [20]. Data analysis was conducted as previously described [21]. After alignment, sgRNAs with zero reads in the control samples were removed. Only genes with over 3 sgRNAs were included in the analysis. The normalization of the count table was relative to the total number of reads for each time point, and replicates were averaged. Log-fold-change (logFC) value for each gene was calculated as the average log2 ratios between the sgRNAs of the 4 treated samples (2 ATMi concentrations at 2 time points each) and the control population. Statistical significance was calculated by the Kolmogrov–Smirnov test for two samples, by comparing the sgRNA population of each gene to the distribution of all sgRNAs from the same condition. Pathway enrichment and functional analyses were conducted using the GSEA [22,23], STRING [24], and Kyoto Encyclopedia of Genes and Genomes (KEGG) databases [25,26,27].

### 2.6. DNA Extraction and High Throughput DNA Sequencing

Genomic DNA was extracted using the Blood & Cell Culture DNA Midi Kit (Qiagen, Hilden, Germany) according to the manufacturer’s instructions. Regions containing the single guide RNA (sgRNA) integration were amplified with the primers specified in key resources table (Appendix A). A Nextera DNA library was generated as previously described [20] and sequenced using NextSeq 500 (Ilumina, San Diego, CA, USA).

### 2.7. Enrichment of Haploid ESCs

h-pES10 cells were washed once with phosphate buffered saline (PBS), then trypsinized using TrypLE Select. Cells were centrifuged and resuspended in fresh human ESC medium containing 10 µg/mL Hoechst 33342 (Sigma Aldrich, St. Lois, MO, USA) and incubated at 37 °C for 30 min. Cells were then centrifuged and resuspended in PBS with 10% KSR and 10 µM ROCKi, filtered through a 70 µm cell strainer (Corning, Glendale, AZ, USA), and sorted by DNA content in BD FACS Aria III (BD Biosciences) using a 405 nm laser. Sorted cells were plated in 6-well plates with fresh growth medium containing 10 µM ROCKi.

### 2.8. Generation of Knockout Cell Lines

Two different sgRNAs were used per gene. sgRNA sequences are summarized in Appendix A. The targeting sgRNAs were cloned into the lentiCRISPR v2 or lentiCRISPRv2 neo-lentiviral vectors, as detailed before [28]. An independent control cell line was generated without insertion of sgRNA. Transduction to recently-sorted haploid ESCs (see methods: Enrichment of haploid ESCs) was performed as described [28]. Between 36–48 h after transduction, medium was replaced again with growth medium supplemented with either 0.3 µg/mL puromycin (Sigma Aldrich, St. Lois, MO, USA) or 0.3 µg/mL neomycin (G418, Sigma Aldrich, St. Lois, MO, USA). Cells were kept under antibiotic selection for 7–10 days. Surviving cells were expanded and were used as bulk knockout culture (NF2- or SAV1-knockout cells) or underwent clonal isolation followed by sequencing of the insertion region to validate mutation (ATM-knockout cells). For clonal isolation, 200 or 1000 knockout cells were seeded on 10 cm plates, and after ~5 days, single clones were isolated and transferred into 12-well plates. A few days afterward, clones were transferred into 6-well plates, and after expansion, genomic DNA was extracted using a Blood/Cell DNA Mini Kit (Geneaid, New Taipei City, Taiwan), amplified using primers flanking the sgRNA area as summarized in Appendix A, sequenced using a 3730xl DNA Analyzer with ABI’s data collection, and analyzed with SnapGene software version 6.0.2 [29].

### 2.9. NPC Differentiation

Neural progenitor cells were obtained as established previously [30], with minor adaptations. *ATM*-knockout ESCs were plated on Matrigel-coated 24-well or 96-well plates at densities of 450,000 and 60,000 cells per well, respectively, and cultured as described to generate NPCs (DMEM/F-12 medium was replaced by knockout DMEM). After the last day of differentiation, the cells were re-cultured with the same medium supplemented with 0, 10, or 30 µM LATSi for 24 h. A quantity of 100 µL of medium from each well in the 24-well plates was transferred to a new well in a 96-well plate, and the 96-well plates (either growth plates or medium-containing plates) were analyzed using a CellTiter-Glo assay as described above.

### 2.10. RNA Isolation and RNA Sequencing

ATM-knockout h-pES10 clones 1 and 3 were plated on 6-well plates with MEF feeder layers. One day after plating, the medium was replaced with standard medium (as control) or medium containing 10 µM of LATSi for 24 h. Cells were then harvested, total RNA was extracted, and libraries for RNA sequencing were generated and sequenced as detailed before [28]. Samples obtained from clones 1 and 3, under the same conditions, were considered replicates. Total reads were mapped to the human GRCh38 reference genome and to mouse GRCm38 using the STAR package [31]. The XenofilteR [32] package in R was used to filter out mouse-originated reads. Count tables and differential analysis were performed using the EdgeR package in R [33].

## 3. Results

### 3.1. Genome-Wide Screen in Haploid hESCs Reveals Synthetic Viable Genes under ATM Inhibition

In order to identify synthetic viable genes in *ATM* deficiency and to elucidate the compensatory genetic network of *ATM*, we performed a genome-wide screen on our haploid hESCs CRISPR/Cas9 loss-of-function library [20] after specific inhibition of ATM phosphorylation. Inhibition of ATM was achieved by using the small molecule KU-55933, a potent and selective ATM kinase inhibitor [34]. To validate the effectivity of KU-55933 (herein ATMi) on our haploid hESC line, h-pES10, we treated the cells with the radiomimetic drug NCS to induce DNA damage [35] under increasing levels of ATMi. A day after treatment, we performed Western blot analysis to detect the kinase activity of ATM, using a specific antibody directed against the auto-phosphorylated form of ATM (S1981 phosphorylation). Our results demonstrated an efficient reduction in ATM phosphorylation after treatment with as little as 20 µM of ATMi (Figure 1A and Appendix A). Consistent with our previous results [20], which indicated a negative, though not detrimental, growth effect on ATM perturbation, administration of increasing concentrations of ATMi had a deleterious effect on human pluripotent stem cell (hPSCs) survival (Figure 1B).

Next, to gain a deeper understanding of ATM’s neurodegenerative effects and its role in apoptosis, as well as to identify therapeutic targets, we performed a genome-wide screen to identify genes that are synthetically viable with ATM deficiency. For that, we treated our hESCs library of loss-of-function mutants either with 60 µM or 90 µM ATMi or with an equivalent volume of DMSO as a control. It is worth noting that to sufficiently enrich the population with relevant gRNAs, we used toxic ATMi concentrations (Figure 1B), which are higher than the minimum required for ATM inhibition (Figure 1A and Appendix A). Thus, to ensure our results reflected the effect of ATM deficiency, we validated our selected hits using low ATMi concentrations or in ATM-knockout cells (see below). ATMi-treated library cells were harvested after 10 and 20 days, following 2 and 4 rounds of treatment, respectively. Genomic DNA was extracted from harvested cells and the sgRNA regions were amplified and sequenced. LogFC was calculated for each sgRNA as a measure of relative abundance between ATMi-treated and control cells, and the average logFC was calculated per gene (hereafter “logFC value” refers to the average logFC of the gene) (Figure 1C).

The sgRNA count values obtained from ATMi treated cells were notably different from those of control cells, reflecting the magnitude of the effect of ATM dysfunction (Figure 1D). As expected, principal component analysis (PCA) of the count values indicated that the deviation correlates with the increasing concentration of ATMi (Appendix A). The primary focus of our research is the fraction of enriched sgRNAs in cells exposed to ATMi with positive and significant logFC values, representing genes whose disruption increases cell survival when ATM activity is blocked. Such genes are suggested to be synthetic viable in *ATM* malfunction. Forty-nine significantly enriched genes (logFC > 0 and FDR < 0.05; Appendix A) were revealed in our screen, of which 28 were not shown to promote growth of haploid hESCs without ATM inhibition in our previous screen of untreated cells [20] (Figure 1E).

### 3.2. Perturbation of the Hippo Signaling Pathway Promotes Cell Survival upon ATM Inhibition

According to the enrichment analysis performed using the KEGG database [25,26,27], Hippo signaling was the most enriched pathway associated with genes with the highest logFC values. The Hippo pathway is highly conserved across species and is involved in various key processes, such as organ growth, cell fate determination, differentiation, proliferation, and apoptosis [13]. Notably, the Hippo signaling cascade was more enriched than pathways with a prominent and well-established role in growth inhibition, such as apoptosis and p53 signaling pathways (Figure 2A). We also looked at the protein–protein interaction network of these highly enriched genes and performed k-means clustering analysis based on the strength and confidence of the associations between the different proteins. Out of the three functional clusters uncovered by this analysis, one represents the Hippo signaling pathway, and another represents apoptotic genes (Figure 2B).

To better understand the impact of Hippo pathway perturbation on cell survival, we compared the effects of Hippo pathway genes and regulators (obtained from the AmiGO 2 gene-ontology database [36,37]) in our screen to those in the control screen without ATM inhibition [20] (hereafter referred to as the control screen). We ranked all genes in each screen according to their logFC (from highest to lowest) and divided the rank of each gene in the control screen by that in the ATMi screen. A relative score greater than 1 indicates a stronger effect of the gene in the context of ATM inhibition, and vice versa. Out of 18 Hippo pathway genes, 16 are expressed in hESCs, of which 12 had a relative score higher than one (*p* = 0.05; hyper-geometric test), 4 of them exceeding 100 (Figure 2C). This result indicates that inhibition of the Hippo pathway enhances cell growth, especially when ATM kinase is inhibited, suggesting that this pathway may be a potential target for rescuing *ATM*-deficient cells.

### 3.3. Genetic Validation of BAG6, SAV1 and NF2 as Synthetic Viable in ATM Deficiency

To validate the results of our screen, we chose to focus on the three genes with the highest logFC values, which were not significantly enriched and had a much lower rank in the control screen—*BAG6*, *SAV1*, and *TRIP12*. *SAV1* is a key member of the canonical Hippo signaling pathway [38], *BAG6* is a known ATM phosphorylation target and has a role in apoptosis, and *TRIP12* is an E3 ubiquitin ligase involved in the DDR, which had a negative logFC in the control screen [39,40]. To this list we added *NF2*, a Hippo pathway regulator [11] which was the second hit in our screen, but was also enriched to some extent in the control screen (Figure 3A).

We employed a validation strategy of CRISPR/Cas9-mediated knockout of the gene of interest, using two distinct sgRNAs simultaneously, followed by survival analysis of knockouts and control cells when exposed to variable ATMi concentrations. In case our gene of interest is synthetic viable with *ATM* loss-of-function, we anticipate a shift rightwards in the survival curve, indicating improved survival of the knockout cells. Indeed, cells with mutated SAV1 and NF2, which are associated with the Hippo signaling pathway, survived significantly better under ATM inhibition than control cells (Figure 3B,C), reinforcing our previous observation that the Hippo pathway is associated with *ATM*. *BAG6*-knockout cells also followed the same trend. Contrastingly, the survival curve of the *TRIP12*-knockout cells did not deviate significantly from the control curve (Figure 3B). In light of the above, we decided to focus on Hippo pathway genes for further investigation.

To ensure that the effect on cell growth was caused by ATM loss-of-function and not due to any other side effects of the drug, we employed yet another genetic approach. We generated three clones of *ATM*-knockout h-pES10 cells, by simultaneously infecting 2 *ATM*-targeting sgRNAs (Appendix A). On the background of *ATM*-knockout clone number 2 (protein depletion shown in Figure 3D), we induced mutations in either *NF2* or *SAV1*. We then compared the growth rate of the double knockout cells to that of single knockout or wildtype cells using a CellTiter-Glo assay. The results revealed that *ATM* mutation alone has a negative growth effect, which is consistent with ATMi treatment results. This effect was reversed by mutations in both *SAV1* and *NF2*, which conferred a significantly higher growth advantage to *ATM*-deficient cells relative to wildtype cells (Figure 3E, left). Since ATM is a key regulator of the cellular response to DSBs, we also analyzed the effect in cells 2 days after DSB induction. As expected, *ATM*-knockout cells were highly sensitive to DSB induction compared to WT cells (Figure 3E, right, ct bars). Notably, the hyper-sensitivity of ATM-knockout cells disappeared when the cells also lacked functional *NF2* or *SAV1* (Figure 3E, right).

### 3.4. Chemical Inhibition of the Hippo Pathway Promotes Growth of ATM-Knockout hESCs and of Their Derived NPCs

Considering the above results, we were encouraged to investigate the possibility of chemically inhibiting the Hippo signaling pathway as a means of rescuing *ATM*-related cell death in the clinic. We examined the effect of CGP3466B (also known as Omigapil maleate or TCH346) [41], which inhibits glyceraldehyde-3-phosphate dehydrogenase (GAPDH) nitrosylation and has anti-apoptotic and neuroprotective effects. CGP3466B is under clinical investigation for the treatment of congenital muscular dystrophy [42] and has also been suggested for the treatment of multiple sclerosis [43]. CGP3466B provides a neuroprotective effect after traumatic brain injury in rats. This effect is mediated by downregulation of the Hippo signaling pathway [44]. Therefore, this chemical is suitable for our purposes both from the molecular and from the clinical point of view. We exposed control and *ATM*-knockout cells to increasing concentrations of CGP3466B and, 6 days after exposure, compared their survival to that of untreated cells. *ATM*-knockout cells showed gradual improvement in survival rates with an increase in drug concentration. Conversely, no such effect was detected in control cells (Figure 4A). These findings suggest that in ATM-knockout downregulation of the Hippo pathway without external stress promotes cell growth. Furthermore, these results indicate that the protective effect of the drug is more apparent in ATM-knockout cells than in control cells, and they are consistent with the positive growth effect of inhibition of the Hippo pathway in *ATM*-deficient cells.

To ensure that our results were caused directly by inhibition of the Hippo pathway, we investigated the effect of Lats-IN-1, an inhibitor of the canonical Hippo pathway kinases *LATS1* and *LATS2*, which are responsible for *YAP1* phosphorylation and restriction [45]. We exposed control and *ATM*-knockout cells to increasing concentrations of the drug and tested their relative survival. At 24 h post-treatment, we detected a clear rise in the viability of *ATM*-knockout cells, with a peak at 10 µM LATSi, as opposed to control cells, which remained largely unaffected (Figure 4B). Based on our analysis, both Hippo pathway inhibitors increased the survival of *ATM*-knockout cells significantly more than that of control cells, reinforcing the potential of Hippo pathway inhibition to rescue *ATM*-depleted cells.

A–T-related cell death is manifested in vivo primarily in nerve cells. Therefore, to further validate the therapeutic relevance of our findings, we tested the effect of Hippo inhibition also on cells that are committed to the neural lineage. For that, we differentiated the pluripotent *ATM*-knockout cells into neural progenitor cells (NPCs), then exposed them to 10 µM or 30 µM of LATSi for 24 h. The viability of treated cells was significantly higher compared to those left untreated (Figure 4C). This result supports the potential of the Hippo pathway as a therapeutic target for the treatment of neurodegeneration in A–T patients.

### 3.5. Mechanistic Insights into the Functional Relationships between ATM and the Hippo Signaling Pathway

In order to further support the clinical potential of LATSi for the treatment of neurodegeneration seen in A–T patients, and to better understand the genetic networks affected by Hippo pathway inhibition in ATM-knockout cells, we performed transcriptome analysis of two ATM-knockout clones with or without the administration of 10 µM LATSi for 24 h. The analysis focused on a list of YAP target genes, obtained from Chu Zhu et al., 2015 [46]. Differential expression analysis revealed elevated expression of most targets of YAP transcription factor, particularly those related to the YAP/TEADs complex, which is associated with cell proliferation. Genes associated with stemness and differentiation, such as POU5F1 (OCT-4), SOX2, and SOX9, were unaffected or somewhat downregulated (Figure 5A). The elevated expression of anti-apoptotic genes is strengthened by a hypergeometric test (*p* = 4.7 × 10^−5^). Together, our results support the conclusion that the Hippo pathway grants synthetic viability to ATM deficiency and indicate a major apoptotic and anti-proliferative role of the Hippo pathway in ATM-depleted cells.

## 4. Discussion

The *ATM* gene has been the subject of intensive research in recent years. Aside from its well-known roles in DDR, ATM also plays various cytoplasmic roles in redox homeostasis, autophagy, insulin metabolism, and more [5,7,10,47]. This indicates that ATM appears to play much more diverse and complex roles in cellular homeostasis and physiology than previously thought. In particular, the relationship between *ATM* gene loss-of-function and the death of neurons in the cerebellum, characteristic of A–T patients, remains largely controversial [4,47,48,49]. This dispute is a major barrier to the development of treatments for cerebellar atrophy, one of the most severe symptoms of A–T syndrome. A better understanding of the molecular pathways affected by ATM may enable the discovery of new treatments or therapies to slow or stop the progression of this debilitating neurological disorder.

Notably, mutations in genes associated with DDR are commonly associated with neurodegenerative syndromes. Examples include NBS1 and MRE11, which are both involved in DSB repair and are associated with Nijmegen breakage syndrome and ataxia–telangiectasia-like disorder, respectively [50]. These findings suggest a link between DDR and the development of neurodegenerative syndromes, and imply that the role of ATM as an effector of the response to DSBs stands at the basis of A–T related neurodegeneration. However, there are multiple evidence for the involvement of oxidative stress in neurodegeneration [51], implying an alternative underlying mechanism for the neurodegeneration seen in A–T patients.

Mechanistic understanding or identification of an effective treatment will benefit patients and enrich our understanding of ATM function and the neurodegenerative process. However, finding a proper model for A–T is a challenging task. Due to ATM’s major role in DNA integrity maintenance, cells obtained from A–T patients are typically chromosomally unstable [52,53]. This makes induced pluripotent stem cells (iPSCs) difficult to obtain and also a problematic model with regard to their genetic background. In vivo mouse models for A–T are also not ideal, since *ATM*-depleted mice do not recapitulate the cerebellar atrophy phenotype [54]. Many attempts to create adequate in vivo models were made over the years but currently only a combination of several mutations led to the characteristic ataxia of A–T syndrome [55,56,57]. We recently demonstrated that some mutations that cause neurological symptoms have detectable growth effects already in the pluripotent state [30]. Modeling A–T in hESCs to investigate the relationships between *ATM* and neurodegeneration is highly advantageous. These cells are genetically intact apart from the induced *ATM* mutation. In addition, hESCs can differentiate towards the neuronal lineage, enabling investigation of ATM role in different stages of neuronal development. According to our previous loss-of-function screen [20], although *ATM* is not essential for normal growth of hESCs, its perturbation does have a negative growth effect at this stage.

In this study, we inhibited the function of ATM kinase using the background of a genome-wide CRISPR/Cas9 library of haploid hESCs. Notably, to obtain the most informative results, we applied a strong selective pressure to enrich for cells that are more resistant to the absence of functional ATM. Therefore, the ATMi concentration used in our screen was higher than the minimum required to inhibit ATM activity. Using high ATMi concentrations can increase the non-specific effects of the drug. Hence, we validated the screen results either using low ATMi concentrations or in ATM-knockout cells. Mutations that were shown to confer high resistance to ATM inhibition were defined as ‘synthetic viable’ with ATM malfunction and considered as potential targets for the rescue of ATM-depleted cells.

The results of our study indicate that mutations in genes involved in the Hippo signaling cascade provide a pronounced growth advantage to cells with deficient ATM activity. Hippo signaling was the strongest hit in KEGG pathway analysis, and a broad investigation of related genes and regulators exhibited high specificity of its effect on cells with inhibited ATM phosphorylation. This growth advantage was detectable already with low concentrations of ATMi and was validated also on *ATM*-knockout clones. Moreover, chemical inhibition of the Hippo pathway by several small molecules, one of which was previously shown to be brain-penetrant [58] and safe for humans [59,60] in clinical trials, increased the survival of ATM-depleted cells more than that of control cells. Importantly, a high increase in cell survival upon Hippo pathway inhibition was shown also in hESC-derived neural progenitor cells, providing valuable insight into the neurological effect of such chemical intervention. Based on the above results we suggest the Hippo pathway as a potential target for the treatment of A–T related neurodegeneration.

Interestingly, Hippo pathway involvement in neurodegenerative processes has been documented in recent years in Alzheimer’s disease, amyotrophic lateral sclerosis (ALS), Huntington’s disease, and retinal degeneration [61,62]. However, the mechanism and extent of Hippo pathway involvement remain unknown. Moreover, in several models of neurodegenerative disorders, pharmacological inhibition of the Hippo pathway was found to be neuroprotective [63,64]. For instance, CGP3466B, which we used here, exerts neuroprotective effect after traumatic brain injury in mice, and this effect is mediated by inhibition of the Hippo pathway [44]. This reinforces the relevance of the Hippo pathway to neurodegeneration in general and strengthens its clinical potential. Nevertheless, our results imply a specific effect of the Hippo pathway in the context of *ATM* deficiency, indicating that the mechanism by which the Hippo pathway is involved in the neurodegenerative process, or the extent of its effect, differ between A–T and other neurodegenerative conditions.

*ATM* perturbation leads to the accumulation of unrepaired DNA damage and to elevated levels of ROS [65,66]. As expected considering ATM’s central role in the DDR, cells derived from A–T patients harbor numerous background mutations and are chromosomally unstable [53]. In addition, expression analysis of human cerebellar neurons derived from A–T iPSCs revealed an aberrant oxidative stress regulation [52]. ATM-deficient human glioma cells were shown to have elevated intracellular ROS levels and to be highly sensitive to ionizing radiation and oxidative stress [66]. Notably, DNA damage and oxidative stress are both activating signals for the Hippo pathway [15,46,67]. Upon DNA damage, several components of the Hippo pathway are phosphorylated and activated, leading to an inhibition of cell proliferation induced by YAP1/TEAD. In parallel, the YAP1/p73 apoptotic complex is formed and promotes transcription of apoptotic genes [15,68,69,70,71]. Elevated ROS levels also activate the Hippo pathway in several ways, one of which is through NF2, which activates MST1, leading to YAP phosphorylation and consequently to YAP cytoplasmic sequestration and degradation [67,72,73]. In WT cells, YAP/p73 interaction is also promoted by ATM in response to oxidative stress [67]. Based on the above evidence, we propose a model explaining the connection between *ATM* and the Hippo signaling pathway (Figure 5B). Our transcriptomic analysis supports this model, demonstrating that the proliferative YAP/TEADs complex target genes are upregulated upon LATS1/2 inhibition in *ATM*-depleted cells, implying basal pro-apoptotic activity of the Hippo pathway upon *ATM* deficiency, even in normal, un-stressful conditions (Figure 5A).

Aside from Hippo signaling components, our genome-wide screen highlights additional mutations with protective effect on *ATM*-deficient cells. One example is BCL2-associated athanogene 6 (*BAG6*), which is involved in a number of processes, such as apoptosis regulation, protein quality control and protein degradation [74]. Interestingly, the BAG6 protein is an ATM and ATR phosphorylation target, which is phosphorylated in response to DNA damage and mediates apoptosis through p53 [75]. In addition, BAG6 forms a complex with BRCA1, which promotes homologous recombination-mediated repair [74]. We have validated the synthetic viability relationship between BAG6 and ATM, implying a functional correlation between the two proteins. The interplay between BAG6 and ATM in relation to neurodegeneration should provide a direction for further research.

Importantly, the genes identified in our screen as synthetic viable with *ATM* malfunction are also relevant in the cancerous context. One of the main types of chemotherapeutics is DNA damaging agents (e.g., Bleomycin and Cisplatin) [76,77]. Since ATM is a major regulator of DNA repair, ATM inhibition-based therapy is under clinical trial for cancer treatment in combination with DNA damaging therapies to maximize anti-tumor efficacy [78]. At present, the personalized medicine approach is gaining momentum, and genetic characterization of the mutational profile of individual tumors is becoming standard of care in clinical practice in order to optimize treatment for each patient. Mutations in genes that are synthetic viable with *ATM* loss-of-function can be used as biomarkers for resistance to ATM inhibitors and thus to minimize unnecessary therapeutic intervention and the accompanied collateral damage.

## 5. Conclusions

In conclusion, our screen pointed out genes that are synthetic viable with *ATM* deficiency. Those genes may be potential therapeutic targets for cerebellar atrophy in A–T patients on one hand, as well as biomarkers for cancer resistance to ATM inhibitors on the other. The Hippo signaling pathway is highlighted here as a promising target for attenuating *ATM*-related cell death in neurons.

## Figures and Tables

**Figure 1 cells-12-01503-f001:**
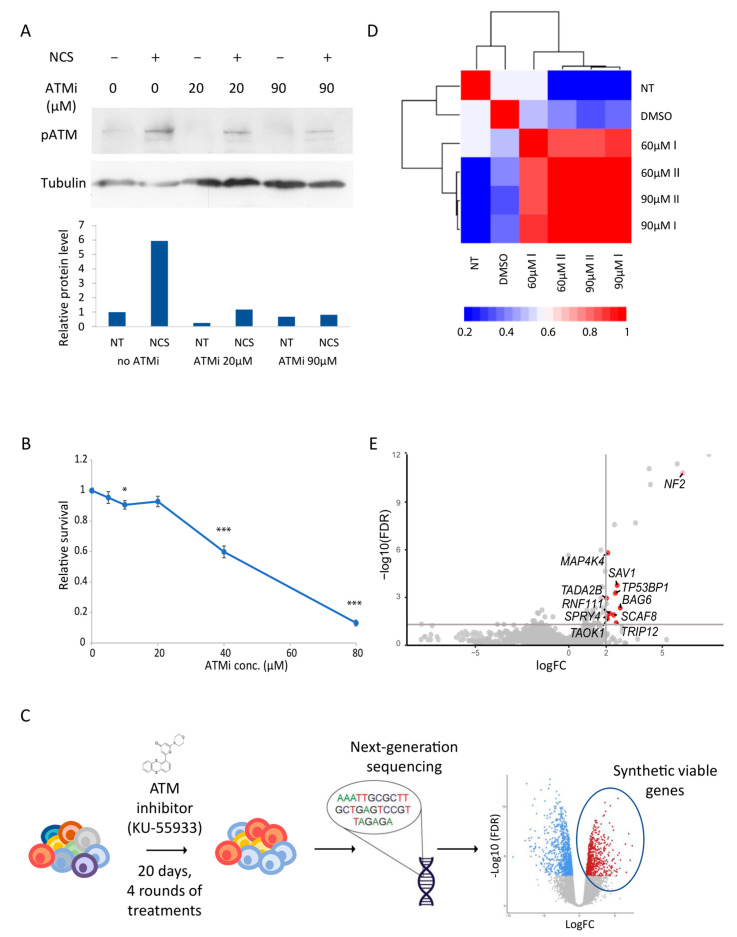
Genome-wide screen in human ESCs reveals synthetic viable interactions with ATM kinase inhibition. (**A**) Western blot analysis of auto-phosphorylated ATM levels in ESCs upon exposure to the ATM inhibitor KU55933 in the indicated concentrations, with or without DNA damage induction by 33 ng/µL neocarzinostatin (NCS), a radiomimetic drug. Quantitative graph of relative protein levels is represented below. Results were normalized to tubulin protein levels as loading control and to no treatment control sample. (**B**) Survival curve representing the relative survival of ESCs following treatment with increasing concentrations of ATMi for 72 h, as estimated by Titer-Glo assay. Each point represents the average of at least 6 biological repeats in 2 independent experiments. Results are normalized to no treatment control. Asterisks represent significance of *t*-test between each data point related to control (* *p* < 0.05, *** *p* < 0.001). (**C**) Schematic illustration of the experimental procedure. (**D**) Heatmap and dendrogram depicting the distance between count values obtained per sgRNA following treatment with different concentrations of ATMi. Correlation levels are represented by different colors. (**E**) Volcano plot indicating significance and log fold change (logFC) of all genes. The top right quarter represents genes with FDR lower than 0.05 and LogFC higher than 2. Labeled in red are genes that were not shown to have a significant positive growth effect in our control screen. NF2 is labeled in light red as it also had a significant positive growth effect in the control screen.

**Figure 2 cells-12-01503-f002:**
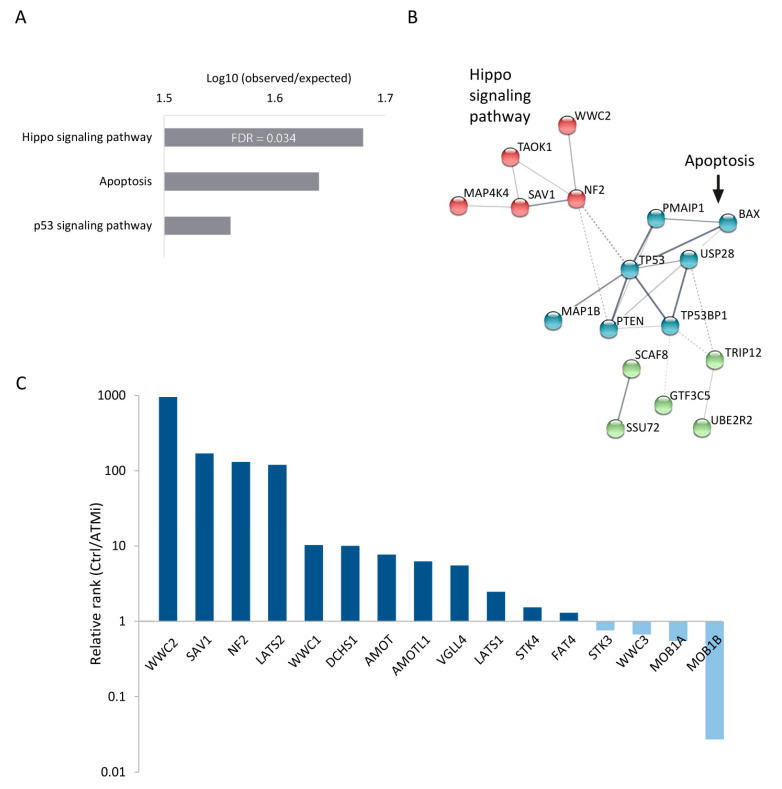
Functional analysis of screen results highlights the Hippo signaling pathway as synthetic viable with ATM dysfunction. (**A**) KEGG pathway analysis of the 30 genes with the highest logFC values. Top 3 results are shown in the graph. (**B**) Protein–protein interaction network of top 30 genes, produced by the STRING database. Interaction confidence is indicated by line thickness; disconnected nodes are not shown. Node colors were determined by k-means clustering algorithm; dotted lines represent edges between clusters. (**C**) Bar plot demonstrating the relative rank of Hippo pathway-related genes and regulators between control and ATMi screen. Genes in each screen were ranked according to their logFC. Y axis is represented by logarithmic scale. Genes with higher growth effect in the ATMi screen are in dark blue; genes with greater effect in the control screen are in bright blue. *p* = 0.05 in hypergeometric test.

**Figure 3 cells-12-01503-f003:**
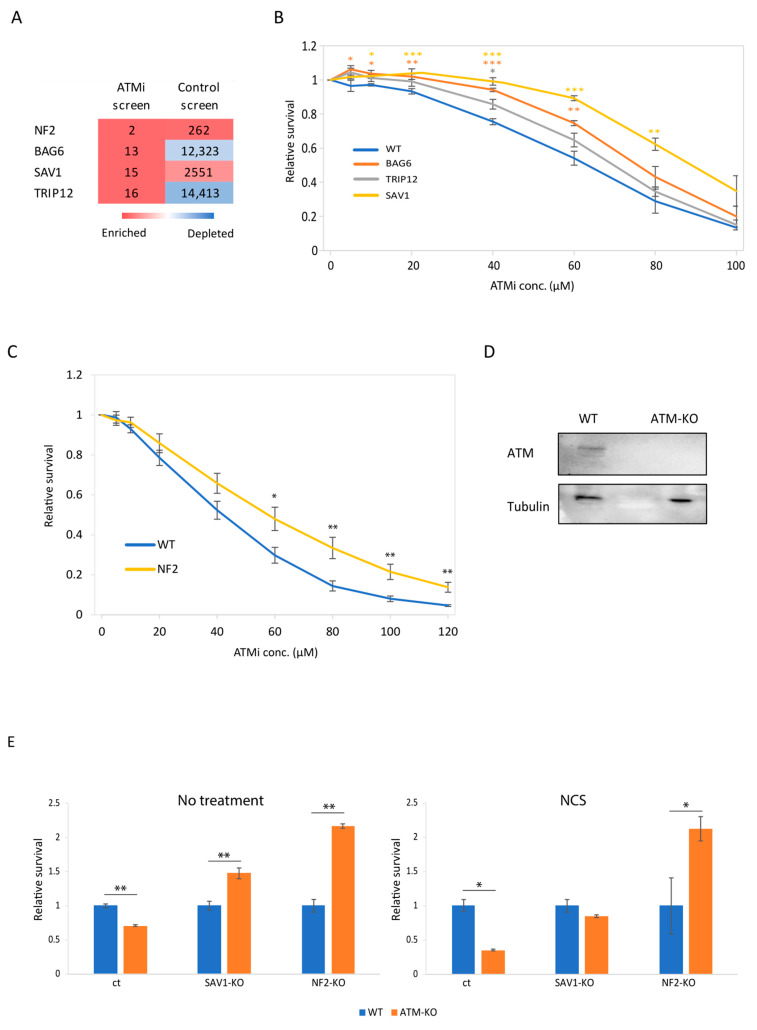
Validation of BAG6, SAV1 and NF2 as synthetic viable genes in *ATM* deficiency. (**A**) Heatmap demonstrating the ranks of the genes chosen for validation in the ATMi and control screens. Color scale ranges from 1 to 16,954. (**B**) Relative survival curve of knockout and wildtype (WT) cells 72–96 h after treatment with ATMi in the indicated concentration. The graph represents the average of 6 biological repeats in 2 independent experiments. Asterisks represent significance of *t*-test between each data point in knockout cells (yellow: SAV1, orange: BAG6, and gray: TRIP12) and WT cells (* *p* < 0.05, ** *p* < 0.01, *** *p* < 0.001). (**C**) Relative survival curve of *NF2*-knockout and WT cells 24 h after treatment with ATMi in the indicated concentration. The graph represents the average of 9 biological repeats in 3 independent experiments. Asterisks represent significance of *t*-test between each data point in knockout and WT cells (* *p* < 0.05, ** *p* < 0.01). (**D**) Western blot analysis of ATM protein levels in *ATM*-knockout clone number 2 compared to WT cells. (**E**) Bar plots indicating the survival of *NF2*- or *SAV1*-knockout cells on the background of *ATM*-knockout clones (double knockouts) compared to the background of WT cells (single knockouts), 46 h after exposure to NCS (right) or without treatment (left). Each point is normalized to the same cell type with WT *ATM* gene. The graph represents the average of 3 biological repeats in 1 experiment. (* *p* < 0.05, ** *p* < 0.01).

**Figure 4 cells-12-01503-f004:**
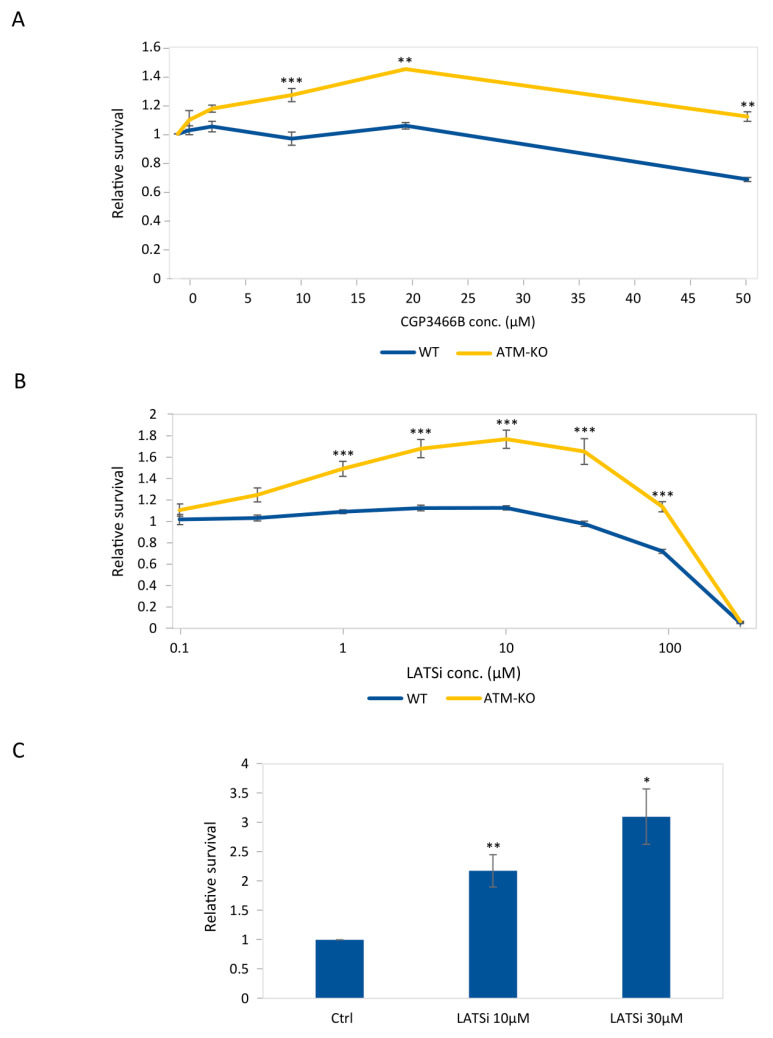
Chemical inhibition of the Hippo pathway promotes growth of *ATM*-knockout hESCs and of their derived NPCs more than growth of WT cells. (**A**) Relative survival of *ATM*-knockout hESC clone and WT cells 6 d after treatment with increasing concentrations of the small molecule CGP3466B. Asterisks represent significance of *t*-tests between each data point in knockout and WT cells. The graph represents the average of 6 biological repeats in 2 independent experiments. (** *p* < 0.01, *** *p* < 0.001). (**B**) Relative survival of *ATM*-knockout hESC clone and WT cells 24 h following treatment with increasing concentrations of LATSi. Asterisks represent significance of *t*-test between each data point in knockout and WT cells. X-axis is represented in a log scale. The graph represents the average of 9 biological repeats in 3 independent experiments. (*** *p* < 0.001). (**C**) Bar plot showing the relative survival of ATM-knockout neural progenitor cells 24 h following treatment with the indicated concentrations of LATSi. Each bar represents the average of 3–6 biological repeats in 1–2 independent experiments (* *p* < 0.05, ** *p* < 0.01).

**Figure 5 cells-12-01503-f005:**
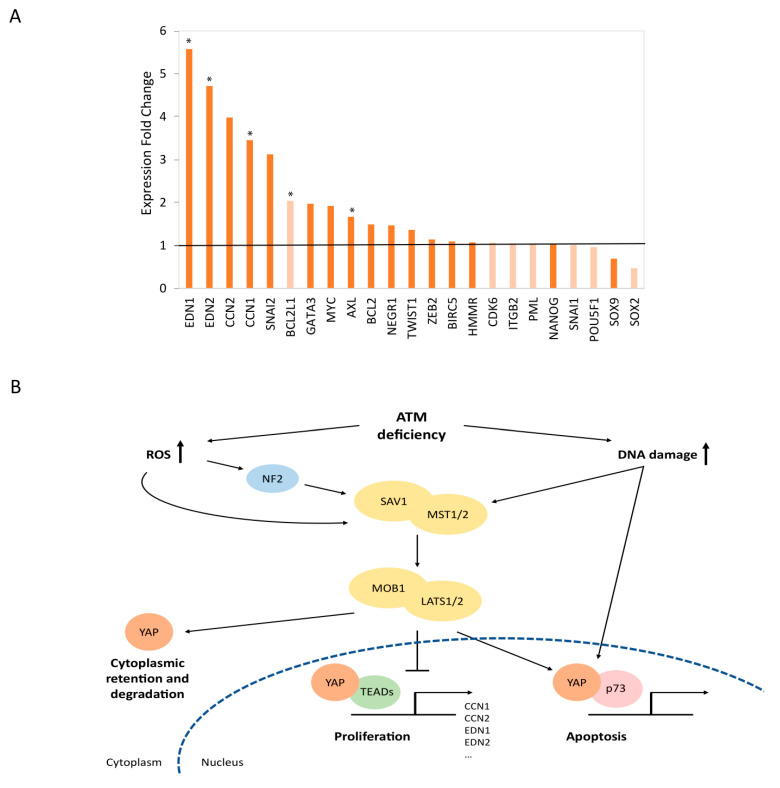
Mechanistic insights regarding the relationships between *ATM* and the Hippo signaling pathway. (**A**) Transcription fold-change of YAP target genes in *ATM*-knockout cells 24 h after exposure to 10 µM of LATSi, compared to no treatment control, as obtained from RNA-sequencing. Dark orange: YAP/TEADs targets. Light orange: other targets of the YAP transcription factor. Asterisks represent significance of differential expression analysis (*p* < 0.05). *p* = 4.7 × 10^−5^ in hypergeometric test. (**B**) Schematic model describing functional relationships between *ATM*-loss and the Hippo signaling pathway.

## Data Availability

RNA-sequencing raw data generated during the current study is available at ArrayExpress under the accession number: E-MTAB-12600.

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
