# Peer review of "Genome-Wide Screening in Human Embryonic Stem Cells Highlights the Hippo Signaling Pathway as Granting Synthetic Viability in ATM Deficiency"

_cells, 2023, doi:10.3390/cells12111503_

Round 1
Reviewer 1 Report
This manuscript entitled "Genome-wide screening in human embryonic stem cells highlights the hippo signaling pathway as granting synthetic viability in ATM deficiency" by Viner-Breuer et al. aimed to screen out synthetic viability in ATM deficient hES and NPCs through genome-wide CRISPR screening. With those over-presented hits, the authors found hippo signaling pathway that might work as synthetic viable with ATM deficiency, which was further validated by CRISPR knockout of ATM along with knockout those enriched hit candidates as well as two hippo signaling pathway inhibitors. Finally, the authors performed RNA-seq trying to provide a mechanistic insight into the functional relationship between ATM and the hippo signaling pathway.
However, the last part of the mechanistic insights into the functional relationship between ATM and the hippo signaling pathway was a little weak.
Overall, the manuscript was well-writing with a logic flow. The experiments are well designed. Most of the data are nice, clean, well-controlled, and organized in a way to support the main conclusion. Figures are labeled clearly. The manuscript is of enough general interest for publication in Cells. The resolution of the figures is strongly suggested to increase for visually better graphics.
Majors:
1. Lane 279, how could it be possible for the authors to use the parameter “LogFC>0 and FDR <0.05” to get 49 significantly enriched genes, please correct the parameters.
2. Figure 4A, the ending point for both conditions with a survival rate greater than 70%, it might be better to use higher concentration of CGP3466B to reduce the survival rate low than 50%.
3. There is no error bar for Figure 5A, it’s not clear how many replicates the authors performed for RNA-seq. And it’s better to validate these results with either RT-qPCR or Western blot.
4. The RNA-seq analysis was a little weak. Please perform more RNA-seq analysis to support the mechanistic insights into the functional relationship between ATM and the hippo signaling pathway.
Minors:
1. Figure 1, quantification of the protein level, there is no error bar for it with indication of replicates.
2. Lane 34. Is reference 1 the end of paragraph 1 in the introduction?
3. Tables S1 are S2 are missing in the submission, please provide those files.
Author Response
We thank the Reviewer for the comments:
Majors:
- Lane 279, how could it be possible for the authors to use the parameter “LogFC>0 and FDR <0.05” to get 49 significantly enriched genes, please correct the parameters.
In this work we conducted a positive selection screen for identifying over-represented sgRNAs, following treatment with ATMi (represented by LogFC>0). At the concentrations selected, ATMi causes depletion of approximately 80 % of the cells on the first round of treatment. Therefore, most of the sgRNAs were depleted from the population. LogFC>0 represent sgRNAs that were enriched rather than depleted. This applies to 36 % of the genes in our experiment, of which only 49 were significantly enriched.
Of note, LogFC and FDR values were computed on 4 replicates, representing 2 ATMi concentrations and 2 rounds of treatment per each. This design reduced the number of significantly enriched genes but allowed us to highlight genes that were over-represented under all conditions.
We now include a Supplementary Table 3 with the significantly enriched genes, including their p-values and FDRs.
- Figure 4A, the ending point for both conditions with a survival rate greater than 70%, it might be better to use higher concentration of CGP3466B to reduce the survival rate low than 50%.
The Hippo pathway plays a role in apoptosis. Since CGP3466B is an anti-apoptotic drug, decreased survival rates are not to be expected. As the Hippo pathway is not activated in un-stressed conditions, such as in Figure 4A, we do not expect to see a reduction in cell growth in control cells following CGP3466B induction. However, we show here that the drug does affect the growth of ATM-KO cells, indicating that in these cells inhibition of the Hippo pathway promotes cell growth even without external stress, presumably because these cells are internally stressed. This is now better explained in the revised manuscript (see page 15 (or page 12 with "no markup" in track changes)).
- There is no error bar for Figure 5A, it’s not clear how many replicates the authors performed for RNA-seq. And it’s better to validate these results with either RT-qPCR or Western blot.
RNA-seq was performed on two different ATM-KO clones in order to increase statistical power. For each clone the experiments were done with or without LATSi. This information is clarified in the new version of the manuscript (see page 5). We also added to the Methods section "Samples obtained from clones 1 and 3, under the same conditions, were considered as replicates" (see page 5).
Further support for the RNA-Seq results is provided in the revised manuscript by a hypergeometric test, which indicates a significant enrichment for genes associated with proliferation (YAP/TEAD targets; P=4.7e-05). This is added to Figure 5A in the revised manuscript.
In addition, we added asterisks representing p-values of the differential expression analyses to Figure 5A.
We believe that these additions support our conclusions even without RT-qPCR, which cannot be performed within the time frame provided for revision by the Editor.
- The RNA-seq analysis was a little weak. Please perform more RNA-seq analysis to support the mechanistic insights into the functional relationship between ATM and the hippo signaling pathway.
The RNA-seq analysis was designed to further support the clinical potential of LATS inhibitor for the treatment of neurodegeneration seen in A-T patients, rather than to gain further mechanistic insights into the functional relationship between ATM and the Hippo pathway. This was clarified in the revised manuscript (see pages 19 (or page 14 with "no markup" in track changes)).
.
Minors:
- Figure 1, quantification of the protein level, there is no error bar for it with indication of replicates.
The quantification of Figure 1A is only for the shown blot, which is above the quantification graph. See Supplement Figure 1A for additional repeats.
- Lane 34. Is reference 1 the end of paragraph 1 in the introduction?
In the first paragraph of the Introduction section, we referred twice to Reference 1.
- Tables S1 are S2 are missing in the submission, please provide those files.
We regret that the submission did not include Tables S1 and S2. We will ensure that they are included in the revised manuscript.

Reviewer 2 Report
The manuscript entitled “Genome-Wide Screening in Human Embryonic Stem Cells Highlights the Hippo Signaling Pathway as Granting Synthetic Viability in ATM Deficiency” by Ruth et al. discussed the link between cerebellar atrophy associated with A-T and Hippo pathway. However, in present condition there are some deficiencies of scientific details. It can be improved after major revision. The issues are listed below.
1. The author must do intergroup significance analysis in figures 1 and 2. P value must be check by specific method and included in the manuscript and figures. This data must be from multiple repeats, otherwise it is not recommended to show in the manuscript.
2. The author must include a positive control in earlier figures for comparison. DMSO can be background solvent and cannot replace the significance of positive control.
3. Intergroup comparison for significance of data is very poor in figure 3 and 4. Authors need to show proper comparison by drawing dotted lines of soft line between groups. Also, stars are not properly placed in figures. I recommend readjusting the figure again carefully.
4. The top ten or 5 expressions of genes must be verified by northern blotting and protein level using flowcytometry, western blotting.
No comment
Author Response
We thank the Reviewer for the comments:
- The author must do intergroup significance analysis in figures 1 and 2. P value must be check by specific method and included in the manuscript and figures. This data must be from multiple repeats, otherwise it is not recommended to show in the manuscript.
Please find our response, for each Figure:
Figure 1A: The quantification of Figure 1A is only for the shown blot, which is above the quantification graph. See Supplement Figure 1A for additional repeats.
Figure 1B: Multiple repeats were performed, as indicated in the figure legend. We have added p-values to the figure (see Figure 1B, page 7).
Figure 1D: This heatmap represents correlation between count values obtained under different experimental conditions of our genome-wide screen. Correlation levels are represented by different colors.
Figure 1E: The Y-axis represents the significance (-log10(FDR)).
Figure 2A: Figure has been updated to include the p-value of the Hippo pathway.
Figure 2B: The line thickness indicates the strength of the data support in this schematic STRING analysis.
Figure 2C: This graph represents a comparison between the ranks of genes in two different screens, a control screen, which was conducted previously in the lab (Nat Cell Biol. 2018) and the ATMi screen, done in this study, and each screen includes multiple repeats. We clarified the term ‘control screen’ in the text.
- The author must include a positive control in earlier figures for comparison. DMSO can be background solvent and cannot replace the significance of positive control.
DMSO was used as a negative control alongside untreated cells. Since we did not expect a particular result, we did not include a positive control in our experimental design.
- Intergroup comparison for significance of data is very poor in figure 3 and 4. Authors need to show proper comparison by drawing dotted lines of soft line between groups. Also, stars are not properly placed in figures. I recommend readjusting the figure again carefully.
We thank the Reviewer for this comment. We tried adding lines between groups, but since the graph contains several curves of different colors, the added lines made it appear cluttered and unclear. Instead, we added the following explanation to the figure legends:
Figure 3B - We now explain in the Figure’s Legend that the yellow, orange and gray asterisks indicate the significance of the effects of mutations in SAV1, BAG6 and TRIP12, compared to the control, respectively.
The asterisks, which indicate significance, have been adjusted in all other relevant figures.
- The top ten or 5 expressions of genes must be verified by northern blotting and protein level using flowcytometry, western blotting.
Further support for the RNA-Seq results is provided in the revised manuscript by a hypergeometric test, which indicates a significant enrichment for genes associated with proliferation (YAP/TEAD targets; P=4.7e-05). This was added to Figure 5A, in the revised manuscript.
In addition, we added asterisks representing p-values of the differential expression analyses to Figure 5A.
We believe that these additions support our conclusions even without RT-qPCR, which cannot be performed within the time frame provided for revision by the Editor.

Round 2
Reviewer 1 Report
The authors have substantially improve the manuscript. No more comments.
Reviewer 2 Report
I reccomend accepting this manuscript as author already revised it
I reccomend accepting this manuscript as author already revised it